# Teaching Inclusive Thinking through an Embodied Metaphor: A Developmental Study

Pablo Herranz-Hernández [1,]* and María Naranjo-Crespo [2]

1  Facultad de Formación de Profesorado, Universidad Autónoma de Madrid, 28049 Madrid, Spain
2  Facultad de Educación, Universidad Complutense de Madrid, 28040 Madrid, Spain; marnaran@ucm.es
*  Correspondence: pablo.herranz@uam.es

**Abstract:** The studies framed in embodied cognition that analyze the metaphor of temperature and its relationship with the feeling of inclusion or exclusion always do so in the first person. That is, they take the perspective of the protagonist who is made to feel included or excluded to see how it affects his or her body thermally. However, there are no studies in which the participants morally evaluate other protagonists who are the ones who feel the inclusion or exclusion and the temperature, projecting these feelings onto them. This paper analyzes the relationship between terms related to temperature (cold or heat) and the moral judgements made toward a person who helps and another who does not help. All this occurs in a situation in which the person making the judgement is not involved in the first person and has to put themselves in the place of the other. In addition, the possible difference in these judgements is examined by comparing children in the fourth grade of primary school with those in the sixth grade. The results indicate that older children give less extreme responses, but these are more influenced by temperature when it comes to morally judging a lack of help. When the behavior is helpful, they judge it morally the same regardless of temperature. In addition, interaction between the two variables appeared. These results have implications in promoting helping in the classroom in order to promote inclusion and represent a useful and accessible resource for such promotion.

**Keywords:** exclusion; inclusion; moral judgement; teaching thinking; thermal metaphor





## 1. Introduction

The debate about the mutual influences of thought and language is not new. On the one hand, linguistic relativism proposes that language influences our manner of thinking, even its most radical versions. One of the most radical exponents of this perspective was Whorf (1940), for whom linguistic systems affect the classifications and categorizations we make of the world. Without taking such a radical stance, there are positions in favor of the influence of language in thought. For example, one position maintains that relational language favors relational thinking (Gentner 2016; Jamrozik and Gentner 2020). Thus, linguistic differences in terms of spatial terms influence how people think about space (Haun and Rapold 2009; Levinson 2003; Levinson et al. 2002). Numerical language also affects the understanding of numerical notions (Carey 2009; Frank et al. 2008; Gordon 2004; Mix 2002; Pica et al. 2004). On the other side are those who defend the aspects common to all languages, which suggests that the way of conceiving things depends on the use of linguistic expressions. One of these common aspects is the metaphors that abound in different languages, such as blame in terms of dirt, moral outrage as disgust, or social exclusion as pain or temperature (Cuetos et al. 2015). For example, Zhong and Liljenquist (2006) determined that when participants copied a cartoon depicting immoral behavior, they considered cleaning products to be more desirable. Chapman et al. (2009) also observed that participants showed facial expressions of disgust when they participated in a game consisting of distributing coins and in which the adversary took advantage and kept more coins.

These examples are manifestations of embodied cognition, an approach that considers that meaning is grounded in perception, interoception, or motricity (de Vega 2021; Barsalou et al. 2008; García and Ibáñez 2018; Horchak et al. 2014; Omori 2008; Ritchie 2008; Strick and van Soolingen 2017). For example, emotional states can be conceived as embodied movements (Kövecses 2005; Lakoff and Johnson 2003; Zlatev et al. 2012). Another good example of this is the study by Khatin-Zadeh et al. (2023), in which the metaphorical relationship between gestures and hand and head movements with embodied emotions is highlighted. For example, when experiencing disgust, the subjects moved their heads backward.

In short, metaphor helps to think of abstract phenomena in a concrete way. Therefore, metaphor constitutes an ally to help students understand complex concepts such as moral values or inclusive value judgements. Although these can be taught in other ways, they may be more complex. For example, Herranz-Hernández and Sánchez-Beato (2021) found that instruction through the logical operator of inclusive disjunction, as opposed to exclusive disjunction, favors the acquisition of inclusive values. Therefore, metaphor, which allows understanding the abstract in simple—and even embodied—terms, can constitute a facilitating tool for inclusive thoughts, values, or value judgements, since negative value judgements could become the previous step to discrimination (Bobbio 2010). Even metaphors presented with a single word can exert an effect on the opinion or preference towards punitive or preventive measures on crime (Thibodeau and Boroditsky 2013). Hence, we must consider the importance of metaphor in favoring inclusive thoughts or moral values.

Regarding the metaphor of social exclusion in terms of temperature, Asch (1946) proposed that people judge others in a psychological dimension of heat and then postulated that the metaphor of heat was related to one's experience of heat. Subsequently, the idea that the sensation of heat plays an important role in social cognition was established (IJzerman and Semin 2009; Semin and Garrido 2012; Semin and Smith 2008). In this sense, for example, in an initial experiment, Zhong and Leonardelli (2008) asked a group of participants to remember a situation in which they felt included and asked the other group to remember a situation in which they felt excluded. Subsequently, both groups were asked to estimate the temperature of the room. The authors observed that the group that remembered the exclusion situation estimated a temperature that was lower than the estimate of the other group by 5° Celsius. In a second experiment, the authors used a procedure devised by Eisenberger et al. (2003) in which a participant played a virtual game of throwing a ball with two other players. In reality, however, they were not players; a computer program assigned the ball passes to one or the other player. In one condition, the participants received the ball from time to time from the other two participants. In the other condition, exclusion, the other players passed the ball between themselves, and the participant in the experiment did not receive it. Afterward, all of the participants had to judge which drinks—hot or cold—they liked the most by means of a survey. The group subject to exclusion considered hot drinks more desirable. IJzerman et al. (2012) observed that body temperature decreases after a situation of social exclusion and that after this situation, the negative feelings experienced can be eliminated by holding a cup of hot tea. In their first experiment, Williams and Bargh (2008) also observed that participants who briefly held a cup of hot coffee (versus ice) judged that a target person had a warmer, more attentive, or more caring personality. In a second experiment, they determined that participants who had a hot (versus cold) therapeutic pad were more likely to choose a gift for a friend than for themselves.

In all these studies, we observe how exclusion and temperature are related and how the thermal metaphors of exclusion/inclusion are rooted or incorporated, that is, in line with the embodied cognition approach, which links meaning with motor, expressive, and emotional aspects (de Vega 2021; Fischer and Zwaan 2008; Moreno et al. 2015; Yang et al. 2017). In them, the subjects directly experience the heat or cold, or the situation of exclusion or inclusion, in the first person. Williams and Bargh (2008) judged the personality of another

individual, but the participants themselves nevertheless experienced the thermal sensation. Therefore, it is worth asking whether this relationship also exists when both the thermal component and the inclusion or exclusion component, that is, both terms or domains of the metaphor, are outside the individual. Do the subjects manifest that metaphorical relation, not in the first person this time but projected towards the others?

Furthermore, when we talk about the ability to put oneself in the other's place, both at the empathic level and at the level of the theory of mind, can we truly say that we put ourselves in their place to the extent that we are influenced by temperature—as if we suffered it in the first person—in the moral judgements of others? The question arises whether the relationship found in studies such as Williams and Bargh (2008) between temperature and personality traits such as warmth or coldness attributed to others can be extrapolated to situations in which we put ourselves in the place of the other. Can this happen when neither the person whose actions we morally judge nor the person person receiving those actions is ourselves? There are neurobiological reasons that lead us to think that this could be the case. For example, there is evidence of a possible union between the thermal sensation of touch and feelings of psychological warmth and confidence in the case of attachment (Insel and Young 2001). It has also been determined that the insular cortex is involved in the processing of psychological warmth and the thermal sensation of heat at the physical level (Meyer-Lindenberg 2008). However, it has also become clear that the insula intervenes in some feelings, such as empathy or trust, as well as in social emotions such as shame and guilt. Some neurons of the fronto-insular cortex appear to be specialized in functions of this type (Balter 2007). Eisenberger et al. (2003) and Kross et al. (2007) also determined that the insula is more intensely activated when we are socially excluded than when we are accepted. Williams and Bargh (2008) analyzed the relationship between temperature and the attribution of personality traits such as warmth/coldness towards other people. However, their study did not relate to moral judgements of the behavior of others when they help or do not help, include or exclude. Although there is a study that does relate moral judgements of social exclusion with the perception of human warmth, that relation is based not on linguistic terms but on facial appearance (Rudert et al. 2017). Furthermore, according to de Vega (2021), explaining abstract language requires, among other aspects, extending the idea of corporeality to interpersonal relationships. Hence, it is important to investigate whether temperature, expressed in linguistic terms, has any effect on the moral judgements of human actions.

Therefore, one of the objectives of this study is to precisely verify whether this metaphorical relationship exists when we project moral judgement onto others in situations that differ thermally. Do the participants judge as morally better or worse an agent who helps or fails to help another in situations of cold or heat? In this sense, it is worth asking whether the thermal metaphor influences such moral judgements and whether it influences in the same manner in the face of behaviors or attitudes of help as opposed to situations in which there is no help.

Another objective of this study is to analyze the developmental aspects of this thermal metaphor with regard to the moral judgement of the behavior of others. Specifically, the goal is to determine whether there are differences between children in the fourth and sixth years of primary school. Both would be within the period of Piaget's concrete operations, but the former more at the beginning of that period and the latter closer to the end of it. Regarding the stages of Kohlberg, some of the children would be in the pre-conventional stage (4–10 years) and the rest would be in the conventional stage (10–13 years). The goal is to observe whether there are differences between the youngest and the oldest children regarding how they morally judge other children who help or do not help in hot or cold situations.

Taken together, these objectives, in addition to contributing to the desirable incorporation of metaphor in the classroom (Deckert et al. 2019; Molina Rodelo 2021; Pérez and Civarolo 2020; Willinger et al. 2019), could be aimed at other challenges, such as finding alternative ways to improve the understanding of metaphors, such as multimodal or sen-

sory metaphors (Marulanda-Páez 2022) and the more general challenge of promoting a more inclusive education in which no one is excluded (Echeita 2019) or the more specific challenge of contributing metaphors in the classroom for the sake of greater inclusion (Rebollo et al. 2013).

## 2. Materials and Methods

### 2.1. Participants

A total of 77 students, 34 boys and 43 girls, aged between 8 and 11 years old (M = 9.96, SD = 1.13) participated in this study. The students were in the 4th year of primary school (aged between 8 and 9 years; 34 children) and the 6th grade of primary school (aged between 10 and 11 years; 43 children). The study was carried out after approval by the school management and with the consent of the parents. Participation was voluntary.

### 2.2. Materials and Procedure

A questionnaire was designed with two versions, one for the experimental cold condition and the other for the experimental heat condition. In each age group and academic level, i.e., for both the 4th graders and the 6th graders, the protocols were randomly assigned, either heat or cold, to the students. Thus, in the group of 4th graders, half of them randomly received the cold protocol and the other half received the heat protocol. The same occurred with the group of 6th graders. The protocol received by the children assigned to the cold condition contained a text that appears in the Appendix A. In this protocol, the children were encouraged to read the text and then respond to questions that appeared after the text. In the instructions, they were told that they could read the text as many times as they wanted and look at it again for their answers if they wished. In the text, they were told a brief story in which some children played in a heated room while another child, named Martín, was on the street, becoming cold. Martín went to the window and observed how the other children played inside. One of the children inside, Álex, proposed that Martín be allowed to play inside with them and thus stop being cold. Another child, Iker, said not to let him in.

Next, the participants were urged to think about Álex's attitude and asked how good or bad Álex was. For this, they were asked to score their attitude between 0 (very bad) and 10 (very good).

Immediately after they were asked to think about Iker's attitude, they were also asked how good or bad Iker was on a scale like the previous one.

Both questions, the one regarding Álex's attitude and the one regarding Iker's attitude, were aimed at analyzing whether children are influenced by the metaphor of temperature in the same way when they have to morally judge a helping behavior (Álex) against a behavior of absence of help (Iker).

Because the helping behavior is socially desirable, it is convenient to ensure that helping is always considered desirable by children and that the results of the answers to the previous questions—the first question about Álex's attitude and the second about the attitude of Iker—are a result of age and/or temperature, which are the variables considered in the study, but not to social desirability. For this, a third question was presented to attempt to control this possible effect. This question caused the children to think about Martín, and they were asked whether they would agree or not to help him by letting him come in. They were asked to score on a scale between 0 and 10, like the previous scales, the extent to which they would agree to let Martín come in.

If there are no differences in terms of the age groups or in terms of the cold or heat conditions in response to this third question, then the possible differences regarding the moral judgement of the behavior of another child, which could appear in both the first questions, could not be attributed to the fact that it is more or less desirable to help in the cold than in the heat or vice versa. Since social desirability influences the self, if it is just as desirable to help with cold or heat, then one would not expect differences in judgements regarding the third question. If there are differences in the first two questions based on

temperature or age, we can be more certain that such differences are not because it is more desirable to help when it is cold than when it is hot or vice versa. In this manner, the differences in the moral judgement of the child who helps or of the one who does not help would be due to temperature and/or age with greater guarantees. We would thus dismiss social desirability as a possible strange variable, not merely social desirability but a possible desirability; let us call it thermal. For example, although Spain is a country of thermal contrasts between winter and summer, especially in cities such as Madrid, in which the study was conducted, this is not the case in the Canary Islands. There are very hot cities such as Seville and other cold ones such as Ávila. Nothing ensures that socially and thermally it is equally desirable to help with cold or heat in all those cities. Therefore, it is essential to take care that the participants in the study do not present differences in the degree to which they would help or consider it advisable to help depending on the cold or heat. Hence, we asked the third question about the degree to which they would agree to help Martín. If there are no differences according to age or temperature, then with more guarantees, we could ensure that possible differences that appear in the first two questions are due to the effect of temperature or age alone.

In the experimental heat condition, the protocol (text and questions) was the same as that of the cold condition except that the cold terms, in the text, were replaced by their heat equivalents. In particular, in the street, it is hot instead of cold; inside the premises, they have air conditioning instead of heating; Martín is sweating from heat instead of shivering from the cold; and when Álex proposes that Martín come in and play with them, he should come inside to cool off instead of warming up.

The children were allowed to take as much time as they needed to read and answer the questions and were seated at separate tables so that they could not be influenced by the responses of their classmates.

The design was 2 × 2. One variable was age or educational level (4th or 6th grade) and the other was the variable thermal condition or temperature (the cold condition or heat condition). The latter was manipulated.

## 3. Results

Regarding the scores given by the participating children for Question 1, the results are presented in Table 1.

**Table 1.** Average scores of badness–goodness for Question 1 according to age and temperature.

| Temperature | Age | Score |
|---|---|---|
| Hot | 4th Primary | 9.71 |
| | 6th Primary | 8.75 |
| | Total | 9.18 |
| Cold | 4th Primary | 9.59 |
| | 6th Primary | 9.20 |
| | Total | 9.37 |
| Total | 4th Primary | 9.65 |
| | 6th Primary | 8.98 |
| | Total | 9.27 |

To determine whether there were significant differences, a 2 (heat and cold) × 2 (younger and older) ANOVA was conducted. There were no differences regarding the temperature ($MC = 0.55$, $F$ [1, 76] = 0.76, $p = 0.39$). There were differences with regard to the age variable ($MC = 8.58$, $F$ [1, 76] = 11.77, $p < 0.01$); the children gave higher scores.

The results of the answers to Question 2 are presented in Table 2.

**Table 2.** Average scores of badness–goodness for Question 2 according to age and temperature.

| Temperature | Age | Score |
|---|---|---|
| Hot | 4th Primary | 0.35 |
| | 6th Primary | 2.31 |
| | Total | 1.43 |
| Cold | 4th Primary | 0.18 |
| | 6th Primary | 0.95 |
| | Total | 0.62 |
| Total | 4th Primary | 0.26 |
| | 6th Primary | 1.62 |
| | Total | 1.02 |

To determine whether there were significant differences, a 2 (heat and cold) × 2 (younger and older) ANOVA was conducted. Differences were observed regarding temperature ($MC$ = 11.13; $F$ [1, 76] = 7.32; $p < 0.01$); the child who does not help in the cold condition scores lower. There were also differences regarding age ($MC$ = 35.49, $F$ [1, 76] = 23.33, $p < 0.01$); the younger children gave lower scores to the child who does not help. There was also an interaction effect (temperature × age) that was significant ($MC$ = 6.59, $F$ [1, 76] = 4.33, $p < 0.05$). The differences between scores in hot and cold conditions were more pronounced in the older children.

Regarding the scores given by the children to Question 3, the results are presented in Table 3.

**Table 3.** Average scores of badness–goodness for Question 3 according to age and temperature.

| Temperature | Age | Score |
|---|---|---|
| Hot | 4th Primary | 9.53 |
| | 6th Primary | 9.33 |
| | Total | 9.42 |
| Cold | 4th Primary | 9.41 |
| | 6th Primary | 9.02 |
| | Total | 9.19 |
| Total | 4th Primary | 9.47 |
| | 6th Primary | 9.17 |
| | Total | 9.30 |

To determine whether there were significant differences, a 2 (heat and cold) × 2 (younger and older) ANOVA was conducted. No differences were observed as a function of temperature ($MC$ = 0.88, $F$ [1, 76] = 0.54, $p = 0.46$). There were also no differences with respect to age ($MC$ = 1.61, $F$ [1, 76] = 0.99, $p = 0.32$).

## 4. Discussion

According to the results for Question 1, in which the children were asked to indicate the degree to which they valued the child who helps to be good or bad, no differences were identified with regard to temperature, although there were differences related to age, with the younger children giving higher scores to the child who helps.

Regarding Question 2, in which the children were asked to indicate the degree to which they valued the child who does not help as bad or good, differences appeared in terms of temperature; lower scores were given to the child who does not help in the cold condition. Differences also appeared in terms of age, with the younger children giving the lowest scores to the child who does not help. An interaction effect was also observed: the differences between the heat condition and the cold condition in the badness–goodness scores were more accentuated in the older children.

Regarding Question 3, there were no differences with regard to temperature or age, which allowed us to rule out social desirability or possible thermal desirability as strange

variables and attribute the results of Questions 1 and 2 with greater guarantees for the variables of age and temperature.

These results, taken as a whole, seem to indicate that there is a certain asymmetry regarding the role of temperature in judgements regarding aid or lack of help. In particular, there is no difference in the role of temperature in morally judging the behavior of the child who helps. However, there are differences when judging the child who does not help, with the participants scoring him as the worst when the situation is cold. Helping is always just as good, but not helping is not as bad. It is worse not to help, according to the children, when the situation is cold.

Regarding age, there appears to be a difference between the two age groups because the younger children polarized their answers more towards Question 1, considering it to be better than the older children did. Regarding Question 2, the younger children also gave lower scores to the child who does not help. That is, they considered him more evil. Thus, the younger children submitted more extreme or polarized responses in cases of both cold and heat. For their part, the older children did not give such extreme scores.

Regarding the interaction between age and temperature, for Question 2, the differences between heat and cold in the scores given to the child who does not help are more accentuated in the older children. The younger children did not appear to be as affected by the temperature when making a moral judgement about the child who does not help. However, the older children do because when the weather is cold, they judged the child who does not help to be worse than when the weather is hot. In the case of the child who helps (Question 1), although there were no significant differences, a similar tendency was observed; the younger children scored nearly as high as the child who helps in the hot situation. The older children differentiated more as a function of temperature. In particular, they valued the child who helps in the cold rather than the hot situation more highly, although the difference was slight and not significant. Therefore, the roots of the thermal metaphor at the time of making moral judgements of behaviors of a lack of help in others appeared to evolve little by little and were not well differentiated from the beginning. The differences in the moral judgements regarding the lack of help depending on the temperature increased with the respondents' age. Perhaps the relationship between temperature and moral judgement regarding a lack of help evolves with development, which would be consistent with the approaches of moral development based on stages, such as those of Piaget and Kohlberg.

Ultimately, with age, the children appear to polarize their responses less, but their responses were more influenced by temperature with regard to morally judging a lack of help. The latter would be in line with previously mentioned results such as those of Zhong and Leonardelli (2008) and IJzerman et al. (2012), although in our case they refer to the moral judgements of another person. Moreover, these findings support the embodied cognition approach, which assumes an imbrication of meaning with perceptual-like experiential systems (Barsalou et al. 2008; García and Ibáñez 2018). Specifically, the participants in our study metaphorically link abstract concepts, such as moral judgements attributed to other children, with the embodied or interoceptive background knowledge they already had in their repertoire about heat or the cold.

In addition, despite the fact that metaphorical meaning is abstract, it can activate embodied representations (de Vega 2021). After all, abstract terms are acquired in our social relations and contribute to modifying our social environment (Borghi 2020; Borghi et al. 2019). Thus, since teaching to think inclusively fosters the acquisition of more inclusive values (Herranz-Hernández and Sánchez-Beato 2021), the thermal metaphor, taught from the corporeal, as highlighted by the present study, represents another way to teach inclusive values.

However, because this study is limited to comparing these two age ranges, it would be interesting to expand it with future studies that consider age groups that are younger and older than those treated here. It could be interesting to analyze what happens to the moral assessment of the helping behavior in more age groups in case there is a similar increase in

the differences between cold and heat conditions, that is, whether the non-significant trend that appeared here could be significant in other studies that contemplate a higher age range. The importance of investigating the evolutionary aspects of metaphor in a more open range of ages lies in the fact that, at a didactic level, the metaphors used should not exceed the complexity or difficulty that each child is capable of interpreting (Marulanda-Páez 2022). In addition to possible age differences in metaphorical comprehension, other factors that facilitate or hinder said comprehension must be taken into account, such as the fact that there is a congruent context with the metaphor in question, especially in interaction with verbal crystallized intelligence (Stamenkovic et al. 2020). In any case, it is important to remember that metaphorical language is economical at the cognitive level; it offers a coherent framework for understanding, often through more accessible images or representations (Banaruee et al. 2019). Factors such as these or perhaps evolutionary ones must be considered in any pedagogical design based on metaphors.

As a primary conclusion of this study, it is worth noting that the temperature metaphor has been determined to affect judgements about the behavior of others when they do not help as well as the developmental aspects of this influence. However, as Barreiro and Castorina (2012) noted, in order to not incur reductionist psychologism when studying the development of moral judgement, one must open up to a relational approach that dialectically integrates the contributions of cognitive development as well as aspects related to collective beliefs and social practices.

Therefore, consistent with this dialectical opening, it is important to emphasize the educational implications of this study, since the thermal metaphor could be used as an element from which to start when teaching values related to aid. These results indicate that the use of situations of cold or heat has different effects on the moral judgement of lack of help. Hence, using one or another temperature at a didactic level could also have differential effects on the teaching of values, so it would have to be considered when teaching values related to inclusion. These educational implications are advantageous since cold and heat are very accessible concepts interoceptively for children, so that they can metaphorize complex concepts such as morals from them. Thus, they can be used at a didactic level in a very simple way. Therefore, this study contributes to the field with the contribution of a simple tool when favoring inclusive values in the classroom. Future research could investigate the role of other embodied emotions (e.g., disgust towards different people) in inclusive values. In addition, future research could increase the age range of this study and, thus, be able to better observe the possible evolution at ages below and above the ages examined here.

**Author Contributions:** Conceptualization, P.H.-H. and M.N.-C.; methodology, P.H.-H. and M.N.-C.; software, P.H.-H.; validation, P.H.-H. and M.N.-C.; formal analysis, P.H.-H.; investigation, P.H.-H. and M.N.-C.; resources, P.H.-H. and M.N.-C.; data curation, P.H.-H. and M.N.-C.; writing—original draft preparation, P.H.-H. and M.N.-C.; writing—review and editing, P.H.-H. and M.N.-C.; visualization, P.H.-H. and M.N.-C.; supervision, P.H.-H. and M.N.-C.; project administration, P.H.-H. and M.N.-C. All authors have read and agreed to the published version of the manuscript.

**Funding:** This research received no external funding.

**Informed Consent Statement:** Informed consent was obtained from all subjects involved in the study. Specifically, as the participants were minors, permission was requested from their parents and the school administration.

**Data Availability Statement:** Data are not available to preserve the privacy of the minors who participated in the study.

**Acknowledgments:** We thank the school management, the students, and their families for their willingness and participation.

**Conflicts of Interest:** The authors declare no conflict of interest.

## Appendix A

Text for the cold condition:

There was once a group of children who were playing indoors. On the street, it was very cold, but inside, they had heating. While playing, the children observed a boy named Martín, who was in the street alone. Martín was shivering from the cold and went to the window of the building. Martín saw how the children inside played. One of the local kids, Álex, proposed that they let Martín in to play with them and thus warm up. But another child, Iker, said not to let him in.

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
