# Peer review of "Teaching Inclusive Thinking through an Embodied Metaphor: A Developmental Study"

_socsci, doi:10.3390/socsci12050267_

Round 1

Reviewer 1 Report

Dear Author/s,

Thank you for the opportunity to review your paper. I have but two suggestions, as follows

(1) include more literature on metaphor and/or language cognition; and
(2) one last round of editing and proofreading (you may want to consider a better word for "awarded" in terms of the scores given by the participants.

I look forward to reading more of your work.

Author Response

Dear reviewer:

First of all, we must thank the work carried out to review our manuscript, as well as the pertinent comments and suggestions made. Next, we respond to them, one by one:

(1) include more literature on metaphor and/or language cognition;

Made. Appears at the bottom of page 1 and top of page 2. It is indicated with the comment tool.

and (2) one last round of editing and proofreading (you may want to consider a better word for "awarded" in terms of the scores given by the participants.

"Given" instead of "awarded". It has been modified in three places. They are indicated with the comment tool.

Sincerely,

the authors

Reviewer 2 Report

Dear Author(s), thank you for submitting your research paper to the journal. I think that your research is valuable and makes important contribution to the discussion about the relationship between metaphors and moral judgement development in children. However, there are a number of points to consider regarding your study.

First, even though the paper is written well, the abstract needs to be re-organised and proofread by an expert. 

Second, the introduction section mostly covers the previous studies but it is also important to provide conceptual understanding of the concepts focused on in your research.  For instance, although, the title of your paper mentions "Inclusive Thinking" and you consider your findings making important contributions to teaching students inclusive values, we do not see much information about these concepts. It is important to explain theoretically how inclusive thinking and other concepts focused on in this study are related to one another. 

Third, I would expect to see a section focusing on the embodied cognition approach and its links with metaphors and the process of making moral judgment. Also, in the discussion, you claim that your findings support the embodied cognition approach, however, it is very less clear how it does. This needs to be elaborated in detail.

In terms of age, to make a statement about "a certain evolution" of children's moral judgements, I think it would have been better to include more age groups in the research design. This is something you also advise for future research studies. 

I hope that my comments will be beneficial in improving your manuscript.

Best wishes,

Author Response

Dear reviewer:

First of all, we must thank the work carried out to review our manuscript, as well as the pertinent comments and suggestions made. Next, we respond to them, one by one:

First, even though the paper is written well, the abstract needs to be re-organised and proofread by an expert. 

It has been conducted. We have rewritten the abstract and we have used the journal´s English editing service.

Second, the introduction section mostly covers the previous studies but it is also important to provide conceptual understanding of the concepts focused on in your research.  For instance, although, the title of your paper mentions "Inclusive Thinking" and you consider your findings making important contributions to teaching students inclusive values, we do not see much information about these concepts. It is important to explain theoretically how inclusive thinking and other concepts focused on in this study are related to one another. 

It is included on page 2, second paragraph.

Third, I would expect to see a section focusing on the embodied cognition approach and its links with metaphors and the process of making moral judgment. Also, in the discussion, you claim that your findings support the embodied cognition approach, however, it is very less clear how it does. This needs to be elaborated in detail.

It is included at the bottom of page 1 and at the top of page 2.

In connection with the comment on the discussion, a clarification to that support has been included in the discussion.

In terms of age, to make a statement about "a certain evolution" of children's moral judgements, I think it would have been better to include more age groups in the research design. This is something you also advise for future research studies. 

The term "a certain evolution" has been replaced by "a difference between the two age groups", a more prudent expression. It was emphasized at the end of the paper that future research could widen the range of ages below and above the ages included in the study.

Sincerely,

the authors

Reviewer 3 Report

My impression is that this paper is a bit of a bait and switch.  While the title promised something about teaching, the study itself is about cold and heat.  Teaching seems entirely absent. 

Moreover, the thermic terms used seem to be about the temperature of characters in scenarios, not the metaphorical terms I would have expected.  For example, we might say that someone who is unfriendly is a "cold" person.  Or someone who is hospitable has a "warm" personality.  The terms used in the article are, on my reading, less clearly moral terms.

Finally, the paper's findings seem relatively inconclusive to me, since " Results indicated that children judge equally bad or good a child who helps, regardless of the temperature terms used "  So regretatably I cannot recommend it for publication in its current form.

Author Response

Dear reviewer:

First of all, we must thank the work carried out to review our manuscript, as well as the pertinent comments and suggestions made. Next, we respond to them, one by one:

My impression is that this paper is a bit of a bait and switch.  While the title promised something about teaching, the study itself is about cold and heat.  Teaching seems entirely absent. 

In our humble opinion, the fact that the use of a term is sufficient for it to have an effect, as occurs in our study or, for example, in that of Thibodeau and Boroditsky (2013), does not eliminate its educational component. It is true that studies can be done in which, seeing the role of the use of thermal terms as occurs here, the educational component is developed or expanded to enhance its effect. But we consider that our study could also be educational, in that sense.

Moreover, the thermic terms used seem to be about the temperature of characters in scenarios, not the metaphorical terms I would have expected.  For example, we might say that someone who is unfriendly is a "cold" person.  Or someone who is hospitable has a "warm" personality.  The terms used in the article are, on my reading, less clearly moral terms.

We consider that, being corporealized metaphors, it is not necessarily a matter of explicitly using moral terms, but simply terms that, when metaphorized in a corporeal way, give rise to different moral judgments. And so they help to conceive a more abstract concept on the basis of something more direct and interoceptively accessible. Moreover, if, as you say, the terms are not clearly moral and yet the effects are observed in the moral judgment, this further corroborates, if possible, their metaphorical role in that judgment.

Finally, the paper's findings seem relatively inconclusive to me, since " Results indicated that children judge equally bad or good a child who helps, regardless of the temperature terms used "  So regretatably I cannot recommend it for publication in its current form.

It is true that when asked about the child who helps, there are no differences, but when asked about when a child does not help another child, there are differences. In addition, an interaction effect between temperature and age is also observed. In our humble opinion, there are many studies published in which no significant differences appear in all the variables studied.

Sincerely,

the authors

Reviewer 4 Report

Dear authors,

I found the underlying idea interesting and innovative. Moreover, it may contribute to the literature related to embodied cognition and teaching-related research. Overall, I enjoyed the novelty of the manuscript. However, there are some issues to be addressed that may help develop the manuscript. 

1.     The abstract needs to be reorganized. It could be more compelling to start by establishing the territory of this research, and then suggest the gap in the literature. Hence, you may clearly state what gap your research is filling. 

2.     In the materials and procedure section, you need to include how many stories were employed in the study and how long they were.

3.     I recommend the language proof of the paper by a native scholar to disambiguate a large number of statements.

4.     The literature needs to be supported by some recent studies as recommended below:

1. On page 7, you discuss ‘The importance of investigating the evolutionary aspects of metaphor’, I suggest that you also add the two works by Stamenković et al. (2020), and Banaruee et al. (2019). You may weigh your discussion by explaining the reasons behind using metaphors.

Dušan Stamenković, Nicholas Ichien & Keith J. Holyoak (2020) Individual Differences in Comprehension of Contextualized Metaphors, Metaphor and Symbol, 35:4, 285-301, DOI: 10.1080/10926488.2020.1821203

Banaruee, H., Khoshsima, H., Zare-Behtash, E., & Yarahmadzehi, N. (2019). Reasons behind using metaphor: A cognitive perspective on metaphor language. Neuro Quantology, 17(3), 108-113

2. Your study investigates the sensing of coldness and heat which are highly related to haptic and interoception feelings. Thus the way that senses or emotions are embodied can provide deep insight for the readers. You may add a highly robust study by Khatin-Zadeh et al. (2023).

Khatin-Zadeh, O., Hu, J., Banaruee, H., & Marmolejo-Ramos, F. (2023). How emotions are metaphorically embodied: measuring hand and head action strengths of typical emotional states, Cognition and Emotion, DOI: 10.1080/02699931.2023.2181314

5.     There are many discrete paragraphs that carry the same idea. You may merge them.

6.     The implications, the study's contribution, and suggestions for further research should be highlighted at the end of the manuscript.

Good luck! 

Author Response

Dear reviewer:

First of all, we must thank the work carried out to review our manuscript, as well as the pertinent comments and suggestions made. Next, we respond to them, one by one:

  1. The abstract needs to be reorganized. It could be more compelling to start by establishing the territory of this research, and then suggest the gap in the literature. Hence, you may clearly state what gap your research is filling. 

It has been conducted. The abstract has been redrafted taking into account your comments. In addition, it has been reviewed by someone from the journal's editing service

  1. In the materials and procedure section, you need to include how many stories were employed in the study and how long they were.

One story was included in the cold condition and another in the hot condition. They were told that they could read it as many times and for as long as they wanted. That is indicated in the materials and procedure section. However, where it says that the condition of heat, the terms of cold were replaced by their equivalents of heat, it has been added that this was in the text, thus reminding the voter that there was only one text for each condition. The order of the terms "cold" and "heat" has also been changed, since they were backwards. A parenthesis has also been added after the word "protocol". Specifically, "(text and questions)", to clarify that it is made up of a text and questions, in order to make it clearer for the reader.

  1. I recommend the language proof of the paper by a native scholar to disambiguate a large number of statements.

It has been reviewed by the journal's translation service.

  1. The literature needs to be supported by some recent studies as recommended below:

  1. On page 7, you discuss ‘The importance of investigating the evolutionary aspects of metaphor’, I suggest that you also add the two works by Stamenković et al. (2020), and Banaruee et al. (2019). You may weigh your discussion by explaining the reasons behind using metaphors.

Dušan Stamenković, Nicholas Ichien & Keith J. Holyoak (2020) Individual Differences in Comprehension of Contextualized Metaphors, Metaphor and Symbol, 35:4, 285-301, DOI: 10.1080/10926488.2020.1821203

It has been included and indicated in the manuscript using the commenting tool, on the right.

Banaruee, H., Khoshsima, H., Zare-Behtash, E., & Yarahmadzehi, N. (2019). Reasons behind using metaphor: A cognitive perspective on metaphor language. Neuro Quantology, 17(3), 108-113

It has been included and indicated in the manuscript using the commenting tool, on the right.

  1. Your study investigates the sensing of coldness and heat which are highly related to haptic and interoception feelings. Thus the way that senses or emotions are embodied can provide deep insight for the readers. You may add a highly robust study by Khatin-Zadeh et al. (2023).

Khatin-Zadeh, O., Hu, J., Banaruee, H., & Marmolejo-Ramos, F. (2023). How emotions are metaphorically embodied: measuring hand and head action strengths of typical emotional states, Cognition and Emotion, DOI: 10.1080/02699931.2023.2181314

It has been included. It appears in red and indicated with a comment.

  1. There are many discrete paragraphs that carry the same idea. You may merge them.

It has been done.

  1. The implications, the study's contribution, and suggestions for further research should be highlighted at the end of the manuscript.

They have been added to the end of the manuscript.

Sincerely,

the authors

Round 2

Reviewer 2 Report

Dear authors, thank you for the improvements done in the manuscript. 

Best

Author Response

Dear reviewer:

Thanks a lot for review our manuscript.

Best Regards

Reviewer 3 Report

The authors have added to the paper in an attempt to show how their already extant work addresses the concerns I proposed earlier.  They do not seem to have addressed them more robustly than adding these clarificatory notes.  Accordingly, I cannot recommend the paper.  A paper allegedly about teaching should substantially address teaching, not merely claim that it would be simple to use concepts in instruction.  

Author Response

Dear reviewer: I appreciate your comments. In this case, it would not be possible, given the deadlines of the journal, to carry out another study with the suggestions that you propose. That would be left for another future investigation in which an educational environment could be designed, with the recommendations that you propose, to evaluate the role of the teaching of the metaphor incorporated in moral judgments. Thanks a lot.
Best regards.